# Prediabetes and Cardiometabolic Risk: The Need for Improved Diagnostic Strategies and Treatment to Prevent Diabetes and Cardiovascular Disease

**DOI:** 10.3390/biomedicines12020363

**Published:** 2024-02-04

**Authors:** Juan Carlos Lizarzaburu-Robles, William H. Herman, Alonso Garro-Mendiola, Alba Galdón Sanz-Pastor, Oscar Lorenzo

**Affiliations:** 1Endocrinology Unit, Hospital Central de la Fuerza Aérea del Perú, 15046 Lima, Peru; alonsogarro2@gmail.com; 2Doctorate Program, Universidad Autónoma de Madrid, 28049 Madrid, Spain; 3Department of Internal Medicine and Epidemiology, University of Michigan, Ann Arbor, MI 48109, USA; wherman@med.umich.edu; 4Hospital General Universitario Gregorio Marañón, 28007 Madrid, Spain; albagaldonsp@gmail.com; 5Laboratory of Diabetes and Vascular Pathology, IIS-Fundación Jiménez Díaz, Universidad Autónoma, 28049 Madrid, Spain; olorenzo@fjd.es; 6Biomedical Research Network on Diabetes and Associated Metabolic Disorders (CIBERDEM), Carlos III National Health Institute, 28029 Madrid, Spain

**Keywords:** prediabetes, cardiovascular risk, type-2 diabetes, plasma glucose, oral glucose tolerance test, impaired fasting glucose, impaired glucose tolerance, hemoglobin A1c

## Abstract

The progression from prediabetes to type-2 diabetes depends on multiple pathophysiological, clinical, and epidemiological factors that generally overlap. Both insulin resistance and decreased insulin secretion are considered to be the main causes. The diagnosis and approach to the prediabetic patient are heterogeneous. There is no agreement on the diagnostic criteria to identify prediabetic subjects or the approach to those with insufficient responses to treatment, with respect to regression to normal glycemic values or the prevention of complications. The stratification of prediabetic patients, considering the indicators of impaired fasting glucose, impaired glucose tolerance, or HbA1c, can help to identify the sub-phenotypes of subjects at risk for T2DM. However, considering other associated risk factors, such as impaired lipid profiles, or risk scores, such as the Finnish Diabetes Risk Score, may improve classification. Nevertheless, we still do not have enough information regarding cardiovascular risk reduction. The sub-phenotyping of subjects with prediabetes may provide an opportunity to improve the screening and management of cardiometabolic risk in subjects with prediabetes.

## 1. Introduction

Prediabetes is a term used to describe individuals whose glucose levels do not meet the criteria for diabetes but who have abnormal carbohydrate metabolism [1]. It is considered an intermediate stage of glucose dysregulation that may precede type-2 diabetes (T2DM) [2]. Prediabetes affected approximately 720 million individuals worldwide in 2021 and will affect an estimated 1 billion people by 2045 according to the International Diabetes Federation (IDF) [3].

Several reports, consensus statements, and guidelines consider prediabetes or intermediate hyperglycemia to be a step in “the natural history of progression from normoglycemia to T2DM” [4,5,6] and state that it should not be viewed as “a clinical entity” [1]. Instead, prediabetes is a condition of high risk for cardiovascular disease (CVD) and other comorbidities. Several epidemiological studies, such as DECODE and the San Antonio Heart Study, have shown that prediabetes is a strong predictor of CVD and that the risk of CVD starts to increase long before the onset of diabetes [7,8]. Prediabetes is also associated with an elevated rate of morbidity and mortality [9]. Recent publications have demonstrated, with moderate certainty, that prediabetes is positively associated with all-cause mortality and the incidence of cardiovascular outcomes including coronary heart disease (CHD) and stroke, as well as with chronic kidney disease, cancer, and dementia [9,10].

Some aspects of the diagnostic criteria for prediabetes are not clear. The different diagnostic criteria proposed, such as the HbA1c, impaired fasting glucose (IFG), and impaired glucose tolerance (IGT) do not identify all subjects at high risk and do not identify the same individuals [11,12]. Moreover, the cut points defined by different organizations are not uniform and may be different for prediabetes diagnosis depending on ethnicity [13]. (Table 1). Several reports demonstrate that the 1 hour-post load glucose level (1 h-PG) during the OGTT is also an early marker for abnormalities in glucose tolerance and identifies at-risk individuals before the traditional 2 h glucose value becomes abnormal [14,15,16,17]. To expand the definition of prediabetes, it may be necessary to individualize or stratify subjects at risk and try to develop criteria to guide intervention [14,15,18]. In this review, we summarize why prediabetes represents a cardiometabolic risk and address the question of how best to treat prediabetes according to the available evidence.

## 2. Cardiovascular Risk, Morbidity, and Mortality in Prediabetes

Prediabetes represents a high-risk condition for the development of T2DM [2,4,5]. However, metabolic abnormalities and some of the long-term complications of diabetes may already exist in some people with prediabetes several years before the onset of T2DM. Prediabetes itself may confer an increased risk of comorbidities and chronic complications that are traditionally viewed as “diabetes-related” [7,9]. 

Prediabetes has been associated with a greater risk for CVD in some, but not all, studies [5]. The data suggest that the association between prediabetes and CVD may only be present in individuals with hypertension [23]. In this context, insulin resistance (IR) appears to be the link between both conditions. Patients with untreated essential hypertension often have IR and compensatory hyperinsulinemia [24]. The coexistence of IR and hypertension can be viewed as a cause–effect relationship, with IR as a cause of hypertension or as a noncausal association [24,25]. On the other hand, IR characterized by defects in insulin-mediated glucose metabolism in tissues, such as the liver, skeletal muscle, and adipose tissue, is one of the earliest manifestations of the constellation of cardiometabolic diseases, including T2DM and CVD [26,27]. A recent publication showed that abnormal glucose tolerance, including prediabetes and diabetes, is frequently present in patients with acute myocardial infarction. These individuals are at a very high risk for recurrent major adverse cardiovascular events (MACE) [28]. Etiologic studies of atherosclerotic cardiovascular disease suggest that IR may be implicated in leading to impaired nitric oxide generation and increased stimulation of the MAP kinase or atherogenic pathway. Insulin resistance and compensatory hyperinsulinemia may also be associated with metabolic syndrome (MetS) [29].

The risk of T2DM and CVD is continuous across the spectrum of IFG, 1 and 2 h- plasma glucose (1 h-PG, 2 h-PG), and HbA1c values. Inevitably, any cut point will be arbitrary. Nevertheless, the goal remains to identify with the greatest accuracy those at risk for developing T2D and CVD [15]. Observational follow-up studies, often more than 20 years in duration, have confirmed the increased risk of mortality from cardiovascular disease and other causes in patients with prediabetes compared to patients with normal glucose tolerance [30,31]. Recent metanalyses performed in the general population have demonstrated that the risk of mortality, diabetes-related complications, and cardiovascular comorbidities are associated with a diagnosis of prediabetes as defined by the American Diabetes Association (ADA). They have concluded, with a moderate level of certainty, that all-cause mortality, CHD, stroke, chronic kidney disease, and dementia are associated with prediabetes [9,10]. It is important to note that the strongest associations were with impaired glucose tolerance as opposed to IFG and HbA1c [9].

On the other hand, an association between 1 h-PG in the OGTT and adverse events, including myocardial infarction, stroke, and all-cause mortality, has also been demonstrated in several longitudinal studies [32]. Specifically, a 1 h-PG ≥ 155 mg/dL (8.6 mmol/L) was associated with progression to T2DM as well as with an increased risk of microvascular disease and mortality in patients with normal fasting plasma glucose (FPG) and 2 h-PG level < 140 mg/dL (7.8 mmol/L) in an OGTT. Moreover, the risk of fatal myocardial infarction or ischemic heart disease was also higher among subjects with elevated 1 h-PG, as were the risks of retinopathy and peripheral vascular complications in a Swedish cohort study [33].

## 3. An Adequate or Insufficient Diagnostic Criterion?

Currently, the diagnosis of prediabetes is established by fasting plasma glucose (FPG), 2 h plasma glucose (2 h-PG) during a 75 g OGTT, or the level of glycated hemoglobin (HbA1c) [4,5]. The cut-point values of 2 h-PG and HbA1c are defined by professional societies such as the ADA, World Health Organization (WHO), and International Expert Committee (IEC) (Table 1); but, in some cases, the values differ according to the scientific organization [1,4,5,34]. Regarding FPG, the ADA and WHO proposed 100 mg/dL and 110 mg/dL, respectively. In the case of HbA1c, the ADA suggests a cut point of 5.7%, and the IEC defines a value of 6.0% as diagnostic of prediabetes. The cut-point criteria, names of the conditions, and the organizations that define them are shown in Table 1. 

### 3.1. Controversy in the Established Cut-Off Values

With respect to the FPG, the ADA recommended lowering the cut point for IFG from 110 to 100 mg/dL in 2003 because this enabled a similar number of people with IFG and IGT to be identified, although it was soon recognized that different people fell into the different diagnostic categories of prediabetes [35]. The WHO did not adopt this new criterion for IFG and continued using 110–125 mg/dL as its diagnostic criterion, perhaps because the evidence that a fasting cutoff of 100 mg/dl was an independent risk factor for CVD was still limited. The cut-off value of 110–125 mg/dL identifies a four-fold increase in the risk of developing diabetes compared with the lower ADA FPG cut point [35,36]. 

Concerning IGT, it has been known that the 2 h-PG, ≥140 but <200 mg/dL, is a strong predictor for future risk of T2DM and CVD. However, it has been suggested that the 1 h-PG in the OGTT is an equal or stronger risk predictor than the 2 h-PG, suggesting that 1 h-PG may replace or complement the 2 h-PG as a convenient marker for prediabetes [14,32]. Several studies have demonstrated that 1 h-PG is a better predictor of dysglycemia than FPG, 2 h-PG, or HbA1c [14,15,37]. Consequently, shortening the standard 75 g OGTT to 1 h may improve its predictive value and clinical utility. 

HbA1c was first considered as a diagnostic criterion for prediabetes in 2009. However, different studies have shown that FPG alone can identify twice as many subjects who progress to T2DM as HbA1c in a range of 5.7–6.4% [38]. Furthermore, low levels of HbA1c (below 6.0%) used as a diagnostic criterion for prediabetes are not as useful or practical, and hence, the cutoff point recommended by the ADA is still controversial. An important systematic review demonstrated that HbA1C values between 6.0 and 6.5% were associated with a substantially increased risk of developing diabetes, with approximately a 25 to 50% incidence over 5 years, while lower values in the range of 5.5 to 6.0% were associated with only a moderately increased risk (less than 25%). The principal limitation of the study was the lack of original data to model the continuous association between HbA1c values and diabetes incidence [39]. The NHANES (2005–2006 and 2011–2014) demonstrated that the prevalence of prediabetes diagnosed with HbA1c 5.7–6.4% was significantly less than the prevalence diagnosed with an OGTT [40]. In general, the major limitation associated with the use of HbA1c includes lower sensitivity than the FPG and OGTT. In addition, its interpretation and accuracy can be affected by the presence of hemoglobin variants (i.e., sickle-cell disease), chronic renal failure, liver disease, iron-deficiency anemia, differences in red-blood-cell lifespan, and age and race [15]. Based on the above, determining the optimal HbA1c range to define prediabetes will be challenging. 

### 3.2. Proposed New Cut-Off Points in the OGTT for the Diagnosis of Prediabetes

Several epidemiological studies have reported that intermediate measurements of plasma glucose during an OGTT are useful for identifying individuals at high risk for the future development of T2DM [19]. A recent report in an Indian population concluded that elevated 30 min-PG is associated with a high risk of incident diabetes after a median follow up of two years, even in individuals classified as having normal glucose tolerance (NGT) by a traditional OGTT [21]. Other reports in Asian people suggest that the 30 min OGTT is useful for identifying populations at high risk of developing T2DM [19,20,21].

As previously mentioned, an elevated 1 h-PG during the OGTT, in addition to being a predictor of T2DM, may be more or at least as important as the 2 h-PG in predicting CVD. The 1 h-PG may also be used to stratify nondiabetic subjects into low-, intermediate-, and high-risk groups, or added to the adult treatment panel (ATP III) criteria for the MetS [14,16]. In the same way, we found that, in subjects with IFG, performing an OGTT and identifying subjects with 1 h glucose levels ≥155 mg/dL, despite a “normal” OGTT, identified individuals who were more likely to have MetS and a worse cardiometabolic risk profile [41]. Different reports have shown that an elevated 1 h-PG value can identify high-risk individuals, and a 1 h-PG value ≥ 155 mg/dL (8.6 mmol/L), despite a “normal” OGTT, can identify those at increased risk for future T2DM [32]. Other reports found that 1 h-PG was a robust predictor of the future risk of T2DM alone or in combination with other metabolic markers, such as α-hydroxybutyrate and linoleoyl-glycerophosphocholine [16,37,42]. 

### 3.3. Other Approaches to Define Patients at Risk

One of the principal reasons to diagnose prediabetes is to identify subjects at risk. However, there are other definitions or scores to evaluate subjects at risk for T2DM and CVD. The MetS, a collection of risk factors that increases the risk of developing heart disease, stroke, and diabetes, and risk scores such as the Finnish Diabetes Risk Score (FINDRISC), the ADA risk score, or the National Institute for Health and Care Excellence (NICE) risk score, may be useful tools to identify subjects at increased risk for diabetes and CVD. 

A problem with MetS and the various risk scores is that they do not identify the same subjects. MetS and prediabetes identify similar populations. However, MetS appears to be more likely to identify those with early renal dysfunction and increased inflammatory activation, while prediabetes is more likely to identify those with early carotid structural changes [43,44]. These findings may be due to the different underlying pathophysiology of these clinical phenotypes in terms of insulin resistance and secretion or to differences in the prevalence of other cardiovascular risk factors [43]. Concerning the scores, such as FINDRISC and other related scores, the principal limitation relates to their application in populations in which they were not initially developed and validated. Limited calibration metrics and a high risk of bias were reported in the performance of FINDRISC in several Latin American countries [45]. 

## 4. An Approach to Risk-Stratify Patients with Prediabetes

Caring for people with T2DM requires individualization of treatment targets and medication regimens. It is also reasonable to individualize treatment for patients with prediabetes. Increased cardiometabolic risk is established by the diagnosis of prediabetes (IFG, IGT, or HbA1c). However, it is not yet clear how the different diagnostic cut offs, pathophysiology, and associated risk factors can be used to individualize treatment.

### 4.1. Diagnostic Criteria and Glycemic Cut-Point Thresholds for Diagnosis

Different diagnostic cut points for prediabetes identify different groups of patients. There is not a clear concordance between the ADA and WHO criteria for prediabetes [46]. An interesting study reports the implications of alternative definitions of prediabetes for the prevalence of prediabetes in U.S. adults defined by IFG, IGT, and HbA1c [47]. The prevalence of prediabetes and the characteristics of the subjects identified by the different criteria differ considerably [47]. A similar conclusion was reached in a recent German study that demonstrated that there is little overlap between people identified with IFG, IGT, and elevated HbA1c [48]. The same study concluded that some factors, such as age, sex, and marital status were associated with only one criterion. According to this, it may be that each cutoff point or diagnostic value for prediabetes may identify a different type of patient with prediabetes.

### 4.2. Pathophysiology According to the Defect in Glycemic Control

The glucose tolerance curve reflects both the defects in insulin secretion and insulin sensitivity that have been described in patients with prediabetes. Some investigators have found that IFG and IGT differ in the mechanisms involved in glucose homeostasis [49]. Thus, the late phase of insulin secretion is more altered in patients with IGT, while the early phase of insulin secretion is more likely to be reduced in patients with both IGT and IFG [50]. Patients with IGT present with marked IR in peripheral tissue, mainly muscle, and mild IR in the liver. In the case of patients with IFG, it is the opposite [51,52]. Moreover, IR is predictive of T2DM and is associated with metabolic disorders under both fasting and postprandial conditions [52]. 

While an OGTT might be a robust clinical and epidemiologic tool to classify risk, it would require conducting OGTTs with at least four glucose measurements (0, 30, 60, and 120 min). This would increase the economic and personal burden and potentially limit its widespread clinical applicability. However, prospective studies may be warranted to evaluate the prognostic utility of the OGTT as a tool for risk stratification and guiding therapy [15].

Several studies have defined the characteristics of individuals with different defects on the OGTT, such as insulin levels or cardiometabolic risk factors, and associated them with defects in β-cell function and/or insulin release [15,18,52]. Others suggest that HbA1c-defined prediabetes is associated with a defective insulin response in combination with inappropriate suppression of glucagon [53]. In the same way, epidemiological studies have attempted to identify phenotypes according to the different cut-off points on the OGTT [14,54,55,56]. The Diabetes Research on Patient Stratification Study (DIRECT), a 48-month follow-up study, showed the profiles of glucose metabolism in different prediabetes phenotypes classified by FPG, 2 h OGTT, HbA1c, and 1 h-PG in the OGTT. The study concluded that, compared to individuals with one defect in the OGTT curve, those with two or three defects showed a higher incidence of T2DM [18]. Also, subjects with 1 h-PG levels ≥ 155 mg/dL and “normal” OGTT were more likely to have MetS and a worse cardiometabolic risk profile [41].

### 4.3. Associated Risk Factors

Overweight and obesity are the most common risk factors associated with the development of T2DM. However, the progressive losses of β-cell mass and function begin in the prediabetic state and are not necessarily associated with obesity [57]. Other risk factors include central obesity, hypertriglyceridemia, and low HDL-cholesterol. Demographic variables, such as age and socioeconomic status, are also important risk factors for the development of T2DM and cardiovascular disease (CVD). The MetS, using the National Cholesterol Education Program ATP III criteria, with the several risk factors described, is considered a prediabetes equivalent by the American Association of Clinical Endocrinology [58]. Nevertheless, the risk associated with the MetS does not exceed its components. Despite this, successful management to prevent T2DM and CVD should address all risk factors involved. Indeed, the MetS is a useful example of the importance of multiple targets for preventive interventions [59].

A recent publication proposed “optimizing strategies to identify high risk of developing T2DM”. In this study, the authors evaluated the conditions IFG, IGT, and HbA1c as criteria for the risk of developing T2DM, considered risk scores for T2DM, including FINDRISC, ADA, and NICE, and found that combining sociodemographic and clinical factors with laboratory tests improves the screening strategy [60]. They also found that FPG was the best of the available laboratory tests when considering issues of feasibility, accuracy, and cost. Moreover, when feasible, including HbA1c, HDL-cholesterol, and triglycerides improved performance, and the two-step strategies, identifying increased (clinical) risk and then assessing for hyperglycemia, reduced laboratory testing with a minimal reduction in the number of future cases detected.

## 5. Established Phenotypes Related to The Development of T2DM

Type 2 diabetes is much more heterogeneous than commonly believed. An interesting publication attempted to reclassify adult-onset diabetes into subgroups according to their outcomes while considering laboratory, autoimmune, and pathophysiological factors, including glutamate decarboxylase antibodies (GAD-A), age at diagnosis, body mass index (BMI), HbA1c, β-cell function, and IR [61]. This framework provides an opportunity to improve the screening for and prevention of diabetes. IR was an important factor for the progression to kidney disease in one of the five groups considered, while the group with insulin deficiency had a high risk for retinopathy. The genetic characteristics associated with the groups differed from the traditional classification. In general, this report represented an important first step towards precision medicine in diabetes [61,62]. 

More recently, a report that focused on participants at increased risk for T2DM (history of prediabetes, family history of T2DM, BMI >  27 kg/m^2^, or history of gestational diabetes), identified six distinct clusters of individuals at risk. Based on metabolic characteristics and risk profiles, the following six sub-phenotypes were identified: cluster 1, individuals at low risk; cluster 2, individuals at very low risk; cluster 3, individuals characterized by β-cell failure; cluster 4, obese individuals at low risk; cluster 5, individuals characterized by IR and fatty liver at high risk; and cluster 6, individuals characterized by a high amount of visceral fat and an increased risk of nephropathy who are at high risk. This proof-of-concept study demonstrated that pathophysiological heterogeneity exists before the diagnosis of T2DM and highlights groups of individuals who have an increased risk of complications, even without rapid progression to overt T2DM [63].

Heterogeneity in the impairment of metabolic parameters suggests that different prediabetes phenotypes may benefit from different treatment approaches. People known to be at risk for T2DM should be tested to identify the presence of all prediabetes defects (IFG, IGT, HbA1c, or elevated 1 h-PG), as they appear relevant for the determination of the actual risk of developing T2DM. This could be the first step in the clinical evaluation. Identifying other associated factors from the clinical history, family history of diabetes, social status, or lifestyle would be necessary to decide on the appropriate path to follow. In Figure 1, we show an approach to evaluate and follow a prediabetic patient. In the figure, we have combined the groups listed above as group 1 (individuals at low risk) and group 2 (individuals at very low risk) into a single group, (cluster 1, individuals at low risk).

It is important to say that neither the publications about novel subgroups of adult-onset diabetes nor the more recent reports about sub-phenotyping of individuals at risk for T2DM evaluated cardiovascular risk or cardiovascular mortality as an outcome [61,63]. On the other hand, in a recent study of a population originating from Ghana, macrovascular complications were evaluated across subgroups of diabetes defined using six clinical and anthropometric variables, and no association with coronary heart disease or stroke was found [64]. Based on the above, a diabetes-risk phenotype that has a greater risk of cardiovascular disease has not yet been defined.

## 6. What Is the Importance of Intervening in Patients with Prediabetes?

Intervention strategies for prediabetes must not only reduce the risk of transition from prediabetes to diabetes but also reduce the risk of macrovascular and microvascular complications. Several studies have evaluated the effectiveness of interventions to delay or prevent progression to T2DM. These studies, including the Da Qing Study, Diabetes Prevention Program, the Finnish Diabetes Prevention Study, the ACT NOW study, and the Indian DPP, have shown a decrease in the incidence of diabetes from about 25 to 72% during lifestyle interventions and with oral drugs (metformin or pioglitazone) at 2, 4, or 6 years [65,66,67,68,69]. Also, the Origin Study demonstrated a reduction in the incidence of diabetes using long-acting insulin (glargine) [70]. Furthermore, they demonstrated that around 20 to 50% of patients achieved normal glucose regulation after the intervention. There are also some studies in which the intervention was weight loss. These studies included Xendos with orlistat, BLOSSOM BLOOM with lorcaserin, a Sequel with topiramate/lorcaserin, a Scale with liraglutide, and a Step program with semaglutide [71,72,73,74,75]. Each demonstrated a reduced incidence of diabetes associated with weight loss. Although successful prevention of any disease requires several essential prerequisites, such as agreed-upon diagnostic criteria, knowledge about risk factors and the natural history of the disease, and affordable and acceptable screening methods to identify high-risk individuals, the randomized controlled trials described above have all demonstrated prevention of T2DM with lifestyle or pharmacologic interventions in high-risk people [76]. 

Two studies deserve special comment. The first is the Da Qing study. This study evaluated 577 patients with glucose intolerance divided into four groups: diet, exercise, both, and a control group. With six years of follow up, all active intervention strategies were effective in reducing the incidence of T2DM [65]. In the observational follow-up study, lifestyle interventions implemented for six years prevented or delayed diabetes for up to 14 years after completion of the intervention. Also, lifestyle intervention led to a reduction in CVD and mortality for up to 23 years [77]. The most recent information from this observational study, presented as “30-year results of the Da Qing Diabetes Prevention Outcome Study” concluded that lifestyle intervention in people with glucose intolerance delayed the onset of T2DM; reduced the incidence of cardiovascular events, microvascular complications, and cardiovascular and all-cause mortality; and increased life expectancy. These findings provide strong justification to continue to implement and expand the use of such interventions to curb the global epidemic of T2DM and its complications [30].

In the Diabetes Prevention Program (DPP), the active interventions were an intensive lifestyle intervention and pharmacological intervention with metformin. In the randomized trial, DPP evaluated 3234 patients over 3 years and demonstrated reductions in the incidence of T2DM of 31% with metformin and 58% with lifestyle [66]. In the first publication during observational follow up, The Diabetes Prevention Program Outcomes Study (DPPOS) concluded that prediabetes is a high-risk state for diabetes, especially in patients who persist with prediabetes despite intensive lifestyle intervention. Reversion to normal glucose regulation, even if transient, was associated with a significantly reduced risk of future diabetes, independent of the previous treatment group [78]. Most recently, complications were evaluated in the DPPOS. These reports concluded that lifestyle intervention or metformin significantly reduced diabetes development [31,68]. There were no overall differences in the aggregate microvascular or macrovascular outcomes between treatment groups; however, those who did not develop diabetes had a lower prevalence of complications than those who developed diabetes [31,68]. This result supports the importance of diabetes prevention. Intervention for prediabetes thus appears to be effective not only during the intervention period but in the long term once the intervention has ended [76,78,79]. Follow up of the DPPOS cohort continues to evaluate other complications, including cancer and dementia. 

It is important to mention that newer drugs, such as sodium–glucose cotransporter 2 inhibitors (SGLT2-Is) and glucagon-like peptide 1 receptor agonists (GLP1-RAs), significantly reduce the risk of CVD in patients with T2DM and CVD, but do not yet have the same level of evidence in nondiabetic patients, including those with prediabetes. Although GLP1s were effective at reducing the incidence of diabetes and in reversing prediabetes to normoglycemia, they may not be the most cost-effective pathway. Their efficacy for weight loss and improving quality of life in nondiabetic patients, including people with prediabetes, has been demonstrated, but there is yet no evidence that they reduce the risk of CVD in the intervention phase or during long-term follow up [80,81]. 

## 7. Conclusions

Cardiometabolic risk is increased in prediabetic states. Refining the definition of prediabetes to include other measures such as components of the MetS might better define individual risk and guide treatment. A first screening with fasting glycemia is reasonable to detect IFG, but a sequential evaluation using the curve of the OGTT might be optimal, especially in subjects at high risk. Individualizing the diagnosis of prediabetes and considering the pathophysiology and clinical characteristics would help to define the risk of both T2DM and CVD and improve the targeting of intervention strategies.

Interventions for prediabetic patients have shown to be of benefit in the short-term in reducing the incidence of T2DM, and, in the long-term, have shown a similar benefit, especially in patients who return to normal glucose regulation. The reduction in the risk of cardiovascular complications is still controversial after intervention in prediabetic patients. However, this does not diminish the importance of lifestyle modification, which should be recommended for prediabetic patients given its other proven benefits.

## Figures and Tables

**Figure 1 biomedicines-12-00363-f001:**
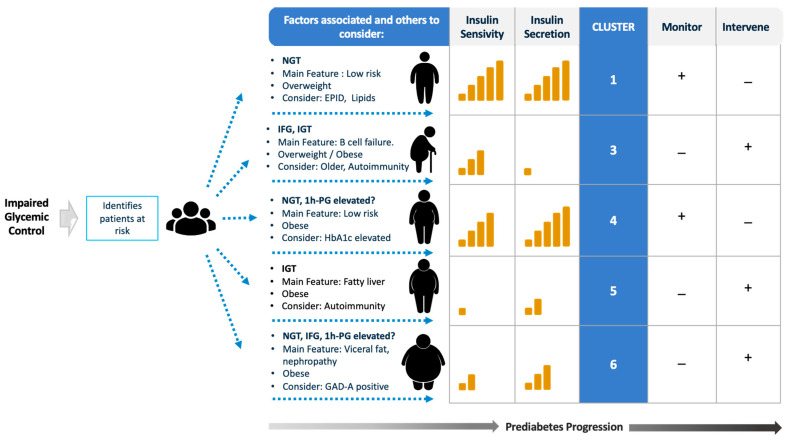
Approach to define the prediabetic patient from screening to the evaluation of risk factors to follow up and intervention. This figure describes possible prediabetes phenotypes based on recent reports [61,62,63,64]. Factors include glycemic parameters, clinical features, impairments in insulin sensitivity and insulin secretion, monitoring, and, finally, intervention. The classification should also weigh exogenous exposure factors, including stress, social/culture, and endocrine disruptors. Cluster 2 was not described individually because it is usually present with normal glucose control and normal weight according to the literature. The "+" symbols emphasizes a primary focus while the "−" suggests a less important action.

**Table 1 biomedicines-12-00363-t001:** Current and proposed diagnostic criteria for prediabetes. Based on American Diabetes Association (2023) [1], World Health Organization (2006) [4] and International Expert Committee (2009) [5]. (*): Not consensus values, range according to the proposed by Hirakawa, Ram, and Hulman [19,20,21]. (**): Cut point proposed for predicting future T2D in several studies by different authors such as Bergman, Abdul, Fiorentino, and Ram [11,14,16,22]. Abbreviations: ADA: American Diabetes Association, WHO: World Health Organization, IEC: International Expert Committee, PG: plasma glucose.

	Time of Measurement	Name of the Hyperglycemic Condition	Organization
**Current Diagnostic Criteria**			
Glucose			
100–125 mg/dL	Fasting	IFG	ADA
110–125 mg/dL	Fasting	IFG	WHO
140–199 mg/dL	2 h-PG (75 g OGTT)	IGT	ADA/WHO
HbA_1C_			
5.7–6.4%	Any time	prediabetes	ADA
6.0–6.4%	Any time	prediabetes	IEC
**Proposed Criteria**			
Glucose			
≥155–182 mg/dL *	30 min-PG (75 g OGTT)	-	-
≥155 mg/dL **	1 h-PG (75 g OGTT)	-	-

## Data Availability

Not applicable.

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
