# Peer review of "Prediabetes and Cardiometabolic Risk: The Need for Improved Diagnostic Strategies and Treatment to Prevent Diabetes and Cardiovascular Disease"

_biomedicines, 2024, doi:10.3390/biomedicines12020363_

Round 1

Reviewer 1 Report

Comments and Suggestions for Authors

The manuscript refers to the important issue however has several flaws.

1.The review is carelessly written and English should be corrected by a native English speaker as the text sometimes is quite difficult to understand. Few examples : Line 49, Table 1, fig 1. Line 191 – the text should be rewritten as it is not clear (MetS or MS ?). Line 239, line 251, line 252 should be HDL-C. Line 301 -lean habitus ? Lines 304-308 – the sentence should be rewritten

2.Line 181 – it is difficult to say that data from 2018 and 2013 are from a recent  study; similarly - line 278 – publication from 2018 ? is not  recent, in my opinion.

3.1hr OGTT was primarily suggested to be performed in children and young adolescents- this could be mentioned by authors.

4.Fig 1 -English to be corrected , the text in the  fig is not clear and full of errors (for ex. Pacient, fat liver ???). Fig. 1 should be better described – it is not enough clear what the black dots mean.

5.Lines 378 and 384 – CVC ? – the abbreviation is not explained earlier , one can only guess it means cardiovascular complications (CVC)

6.References 66 and 68 are not assigned  to the authors  ? N Engl J Med. 2002 Feb 7;346(6):393 ?;  The Lancet Diabetes & Endocrinology. 2015 Nov;3(11):866–75.

7. The text would  benefit if shortened.

Comments on the Quality of English Language

The manuscript refers to the important issue however has several flaws.

1.The review is carelessly written and English should be corrected by a native English speaker as the text sometimes is quite difficult to understand. Few examples : Line 49, Table 1, fig 1. Line 191 – the text should be rewritten as it is not clear (MetS or MS ?). Line 239, line 251, line 252 should be HDL-C. Line 301 -lean habitus ? Lines 304-308 – the sentence should be rewritten

2.Line 181 – it is difficult to say that data from 2018 and 2013 are from a recent  study; similarly - line 278 – publication from 2018 ? is not  recent, in my opinion.

3.1hr OGTT was primarily suggested to be performed in children and young adolescents- this could be mentioned by authors.

4.Fig 1 -English to be corrected , the text in the  fig is not clear and full of errors (for ex. Pacient, fat liver ???). Fig. 1 should be better described – it is not enough clear what the black dots mean ?

Author Response

Reviewer #1

Thanks for ths comments and suggestions.Here we answer point by point to the requirements:

The manuscript refers to the important issue however has several flaws.

1. The review is carelessly written, and English should be corrected by a native English speaker as the text sometimes is quite difficult to understand.

The report has been reviewed again and carefully edited by a native English speaker.

Few examples :

Line 49.

Now lines 50-52. The sentence has been rewritten.

Table 1, fig 1.

Figure 1 has been edited to correct typos and provide further clarification.

Line 191 – the text should be rewritten as it is not clear (MetS or MS ?).

Now line 185. MS has been corrected to MetS.

Line 239.

Now lines 211-213. The text has been rewritten.

Line 251, line 252 should be HDL-C.

Now line 25. For consistency, HDL has been changed to HDL-cholesterol throughout.

Line 301 -lean habitus ?

Now line 265. The phrase has been deleted.

Lines 304-308 – the sentence should be rewritten

Now lines 270-272. The sentence has been rewritten as requested.

2.Line 181 – it is difficult to say that data from 2018 and 2013 are from a recent study;

Now line 163. The phrasing has been changed to delete the term ‘recent’.

Similarly - line 278 – publication from 2018? is not recent, in my opinion.

Now line 243. The phrasing has been changed to delete the term ‘recent’.

3.1hr OGTT was primarily suggested to be performed in children and young adolescents- this could be mentioned by authors.

Answer: The reviewer is correct but the first reference to the 1hr OGTT refers to adults (reference 14).

4.Fig 1 -English to be corrected, the text in the fig is not clear and full of errors (for ex. Pacient, fat liver ???). Fig. 1 should be better described – it is not enough clear what the black dots mean.

Figure 1 has been edited to correct typos and provide further clarification. ‘Fat” has been changed to ‘fatty’ and ‘Pacient’ has been changed to ‘Patient’.

5.Lines 378 and 384 – CVC? – the abbreviation is not explained earlier, one can only guess it means cardiovascular complications (CVC)

Now lines 330-335. The abbreviation has been changed to CVD throughout and clearly defined as referring to cardiovascular disease.

6.References 66 and 68 are not assigned  to the authors  ? N Engl J Med. 2002 Feb 7;346(6):393?;  The Lancet Diabetes & Endocrinology. 2015 Nov;3(11):866–75.

Now references 68 and 70. Authorship has now been added.

7. The text would benefit if shortened.

As requested, we have edited and shortened the text.

Reviewer 2 Report

Comments and Suggestions for Authors this is a very interesting review regarding the association between cardiovascular risk and risk of progression to type 2 diaebetes in subjects with prediabetes.  The topic is not so original and much paper have been released on pre diabetes, however, the problem is still clinically relevant.  The review article is well written and the topic well focused and I don’t have particular issue regarding this manuscript.  English language in clease and figures are interesting. Minor issues need to be addressed Some typo needs to be amended throught the manuscript for example “patient” instead of “patient” in figure 1. I have some concerns regarding the statement in 3.3 section. In fact, several studies conducted in patients with prediabetes demonstrated an alterated inflammatory profile and a early renal impairment which is associated with cardiovascular risk, for example: 10.1016/j.numecd.2021.08.030. In my opinion this sentence should be amended   Furthermore, authors did not report a possible role of early alpha cell dysfunction in this subjects. The review is not focused on physiopathological issue however a paper reporting alpha cell dysfunction in these subjects could be cited 10.1007/s00592-014-0555-5.

Author Response

Reviewer #2

Thanks for ths comments and suggestions.Here we answer point by point to the requirements:

This is a very interesting review regarding the association between cardiovascular risk and risk of progression to type 2 diaebetes in subjects with prediabetes. The topic is not so original and much paper have been released on pre diabetes, however, the problem is still clinically relevant.  The review article is well written and the topic well focused and I don’t have particular issue regarding this manuscript. 

English language in clease and figures are interesting.

Minor issues need to be addressed Some typo needs to be amended throught the manuscript for example “patient” instead of “patient” in figure 1.

Answer: We have carefully edited the manuscript and corrected the typos.

I have some concerns regarding the statement in 3.3 section. In fact, several studies conducted in patients with prediabetes demonstrated an alterated inflammatory profile and a early renal impairment which is associated with cardiovascular risk, for example: 10.1016/j.numecd.2021.08.030. In my opinion this sentence should be amended  

Answer: We have revised the sentence and included the reference as reference 44.

Furthermore, authors did not report a possible role of early alpha cell dysfunction in this subjects. The review is not focused on physiopathological issue however a paper reporting alpha cell dysfunction in these subjects could be cited 10.1007/s00592-014-0555-5.

Answer: We have now cited α-cell dysfunction and cited the reference as reference 53.

Round 2

Reviewer 1 Report

Comments and Suggestions for Authors

The manuscript has been sufficiently corrected and improved  with only one exception : Fig 1 - should be Paients instead of Pacients.

Comments on the Quality of English Language

Fig. 1 - Pacients should be corrected into "patients"

This of course can be changed when proofreading.